# Alcohol Use and the Risk of Colorectal Liver Metastasis: A Systematic Mapping Review

**DOI:** 10.3390/biology12020257

**Published:** 2023-02-06

**Authors:** Roshan Sapkota, Joseph Zakaria, Emily Glenn, Heather Richard, Ahmad Rimawi, Martin Tobi, Benita McVicker

**Affiliations:** 1Department of Internal Medicine, University of Nebraska Medical Center, Omaha, NE 68198, USA; 2McGoogan Health Sciences Library, University of Nebraska Medical Center, Omaha, NE 68198, USA; 3Department of Internal Medicine, Southern Illinois University, Springfield, IL 62702, USA; 4Research and Development Service, John D. Dingell VA Medical Center, Detroit, MI 48201, USA; 5Research Service, VA Nebraska-Western Iowa Health Care System, Omaha, NE 68105, USA

**Keywords:** colorectal carcinoma, alcohol use, colorectal liver metastasis, alcohol-associated liver disease, mapping review

## Abstract

**Simple Summary:**

Colorectal cancer (CRC) is the third most diagnosed cancer worldwide and the second leading cause of death in the United States. Many deaths associated with CRC are due to the metastatic spread of colon tumors to the liver. Drinking alcohol has been associated with liver diseases and an increased risk of developing colon cancer. However, it is not understood whether alcohol consumption and related liver damage associates with liver metastasis and worse outcomes in CRC patients. The objective of this study was to conduct a systematic mapping review of the literature to identify gaps in research investigating alcohol use and the risk of colorectal liver metastasis (CRLM). The review identified 14 publications in the last decade that evaluated alcohol use, the risk of CRLM, and contributing mechanisms involved in promoting metastasis in the alcohol-injured liver. The number of studies identified demonstrates the limited focus in current research evaluating the consequences of alcohol consumption in metastatic disease and the health of CRC patients. Overall, significant gaps in this area of research have been identified demonstrating the need for a better understanding of major worldwide healthcare issues related to alcohol use and CRC outcomes.

**Abstract:**

The consumption of alcohol has long been associated with the development of liver disease as well as cancers including colorectal cancer (CRC). Leading healthcare concerns include the prevalent use of alcohol and the high burden of CRC mortality. Many CRC deaths are attributed to the development of colorectal liver metastasis (CRLM) as the liver is the foremost site of CRC spread. However, an association has not been defined for the role of alcohol intake and related liver injury with the development of CRLM. Here, a mapping review of recent research was undertaken to evaluate the relationship between alcohol consumption and the risk of CRLM. The literature search revealed 14 articles meeting the inclusion criteria that included patient database analyses and preclinical studies. Most of the human data analyses found alcohol use independently associates with worse CRC outcomes. The preclinical evaluations identified several pathways involved in the alcohol-mediated promotion of CRLM burden and CRC cell metastatic behavior. The limited number of studies identified exposes a significant need for more prospective analyses to define the role of alcohol intake and advanced CRC as well as the translation of preclinical research to fully characterize targetable mechanisms for the generation of new therapeutic options.

## 1. Introduction

Colorectal cancer (CRC) ranks as the third most common cancer worldwide and the second leading cause of cancer death in the United States [1]. Mortality is primarily due to the development of colorectal liver metastasis (CRLM), which presents within three years of diagnosis in up to 50% of CRC patients [2]. Therefore, CRLM acts as a significant prognostic factor in patients with CRC. The identification of contributing factors aims to reduce the morbidity and mortality associated with CRLM. However, preventative strategies to mitigate the spread of CRC to the liver are hampered by the lack of defined targetable mechanisms. The contribution of lifestyle habits such as smoking, consumption of red meat, and alcohol use in mechanisms of CRC development have been studied for many years [3,4,5]. Of interest, alcohol consumption is considered a significant targetable factor associated with the adverse outcomes of CRC, yet the role of alcohol intake in liver metastatic disease remains to be characterized.

Alcohol consumption remains deeply embedded within our social fabric. The 2019 National Survey on Drug Use and Health (NSDUH) indicates that 85.6% of people ages 18 and older drank alcohol at some point in their lifetime [6]. Moreover, 25.8% report they have engaged in binge drinking episodes within the month being surveyed [6]. This high prevalence of alcohol intake has continued, especially during the recent COVID-19 pandemic. In the United States, data from a national survey of adults found that excessive drinking increased by 21% during the pandemic [7]. Further, it is projected that the elevated consumption rates since March 2020 will contribute to a significant increase in alcohol-related mortality within the next several decades [7].

The consequences of alcohol use significantly impact public health and the development of diseases including cancers. As reviewed by McNabb et al., multiple studies have linked the aberrant and heavy use of alcohol to the higher incidence of CRC [8]. However, there is conflicting data on the role of alcohol consumption on the risk of colorectal liver metastases. Some reports indicate that alcohol use correlates with increased CRLM identified at initial CRC diagnosis as well as with hepatic metastases that occur over time [9,10]. Further, studies have shown that CRC patients with an alcohol use history often require intensive follow-up due to reduced liver function, unresectable hepatic lesions, and poor survival rates [3,11,12]. However, it has also been shown that light drinking or wine consumption positively associates with colorectal cancer survival [12,13,14]. Thus, the data remain inconclusive to the role of alcohol in colorectal cancer outcomes with clarification needed for the specific association of alcohol use in the risk and progression of CRLM. Such clarification is especially crucial in the United States where drinking alcohol in a chronic and episodic manner is highly prevalent and where the COVID-19 pandemic has drastically increased alcohol consumption patterns [7]. This literature review assessed recent research investigating the relationship of alcohol consumption and the development of advanced disease of colorectal cancer.

## 2. Materials and Methods

### 2.1. Search Strategy

A literature review was conducted to evaluate recent investigations into the relationship of an alcohol use history and/or presence of alcohol-associated liver disease with the development of colorectal liver metastasis. A comprehensive search was conducted in PubMed and Embase for reports published between 2010 and 2021. The search strategy included keywords and controlled vocabulary for alcohol consumption, alcoholic liver disease, liver metastases, colorectal cancer, and prognosis. Full details of the database search strategies can be found in the Appendix A.

### 2.2. Inclusion and Exclusion Criteria

The search strategy and inclusion and exclusion criteria were determined with the guidance and resources of the McGoogan Health Sciences Library at the University of Nebraska Medical Center. The full inclusion and exclusion criteria are listed below (Table 1). Due to the robust correlation between liver metastasis and CRC outcomes, studies that documented CRC mortality were included although these studies lacked the liver metastasis criteria. 

### 2.3. Data Extraction

Three authors (R.S., J.Z., and H.R.) screened the abstracts, reviewed the references of eligible manuscripts, and performed full review and data extraction on selected studies. Extracted data included the study classification (preclinical or clinical), study design (retrospective, prospective, animal, and cell culture), mode of alcohol documentation, preclinical model of ethanol administration, risks associated with alcohol use patterns (heavy, light, or binge drinking), outcomes (risk of CRC mortality or CRLM due to alcohol), and potential mechanisms involved in alcohol-related CRLM. The search and study selection process resulted in the identification of total of 207 studies for abstract review with a total of 14 full-text articles satisfying the criteria for qualitative full manuscript evaluation. Figure 1 is the PRISMA flow chart showing the mapping review selection of manuscripts reporting the association of alcohol in colorectal cancer outcomes including liver metastasis.

## 3. Results

Of the 207 studies meeting the search criteria, less than 10% evaluated the role of alcohol consumption in colorectal cancer outcomes. The 14 included studies were classified as preclinical, clinical, or a combination of both according to respective animal/cell culture or patient database/cohort assessments. The included papers were mostly clinical in nature with 9 of 14 studies involving the analyses of human subject data (Figure 2A). The studies were performed in seven different countries (Australia, Canada, China, Korea, Italy, Germany, and the United States) with a majority of work performed by investigators in North America (Figure 2B). There were three studies involving collaboration between 2 or more countries [16,17,18].

### 3.1. Clinical Studies Evaluating the Role of Alcohol and CRC Outcomes

As outlined in Table 2, nine clinical studies were reported during the last decade consisting of five retrospective reviews and four prospective analyses that included cohort and population-based survey analyses. The majority of subjects were male in all but one study [19]. An association with sex was made in three of the evaluations [19,20,21] and was suggested in another [22]. In two of the reports, females were underrepresented due to low numbers [18] and the significant lack of women who drank alcohol in the study population [19]. 

The sample size varied between the studies with a range of 102 to 173,367 subjects evaluated for alcohol consumption and CRC outcomes. The collection and distribution of alcohol consumption patterns also varied between the studies. Ascertainment methods consisted of retrospective surveys, interviews, questionnaires, and review of Electronic Medical Records. Some studies recorded the alcohol consumed by beverage type and ethanol concentration while others utilized a simple yes-or-no query. When noted, specific intake patterns were provided as quantity (grams/day or week) or frequency (drinking days/week), with subjects characterized as light, moderate, or heavy drinkers. The association of the alcohol intake was reported for digestive cancer mortality [19], all-site cancer deaths [20], or CRC-specific outcomes [9,12,13,16,18,21,22].

Bubble plots were used to review the clinical findings and to identify research gaps. Figure 3 indicates that all but two of the studies demonstrated a negative impact of alcohol on CRC outcomes regardless of how alcohol intake was surveyed or the size of the population studied. The studies that were not able to show a relationship between alcohol consumption and CRC mortality noted that higher alcohol intake trended towards a poorer CRC prognosis [13,16]. Interestingly, when alcohol intake was stratified by beverage type, it was determined that a healthier lifestyle with light alcohol use, specifically wine, associated with a more favorable prognosis [12,16]. The protective effect of wine consumption was lost when higher quantities of alcohol was consumed (Figure 4). All the studies measured prediagnostic alcohol intake using baseline self-reported or self-recalled data. None of the studies assessed whether CRC patients continued to drink following CRC diagnosis.

Figure 5 outlines the overall findings from the clinical studies published during 2010–2021. Overall, the majority of the studies found that alcohol consumption, especially heavy intake (i.e., >300 g/week or >3 drinks/day), associates with worse CRC prognosis and the development of liver metastasis. Two reports were unable to show a relationship between alcohol use and CRC mortality. Interestingly, the consumption of wine was found to associate with better CRC outcomes in some of the studies although the contribution of socioeconomic status and healthy lifestyle factors remains to be defined.

### 3.2. Preclinical Investigations into the Role of Alcohol in Advanced CRC

Since 2010, seven studies have evaluated the effect of alcohol on CRC cell behavior and liver metastasis using preclinical in vitro cell/tissue analyses and in vivo animal models (Table 3). The investigations employed the use of mouse and human cancer cell lines, human tissue, and murine models, with most of the papers incorporating the latter. Two of the studies were follow-up experiments to patient data findings to define contributing prometastatic mechanisms [18,22]. Figure 6 depicts the alcohol exposure models which included the incubation of CRC cell lines with physiological concentrations of ethanol in culture, or in vivo studies with mice fed ethanol in water or in the Lieber–DeCarli (LD) diet [23]. The animal studies incorporated immunocompetent (C57Bl/6) as well as immunodeficient mice (Balb/c, Rag1 knockout strains), and alcohol administered by either mode (water or LD diet). Results from the in vivo assessments indicate that regardless of the different experimental designs, enhanced CRC tumor burden was identified in four independent studies with unique contributing mechanisms identified [17,22,24,25]. Three of the papers evaluated alcohol’s role in facilitating a permissible environment in the liver for CRC cell seeding, dissemination, and growth [24,25,26]. It was determined that alcohol accelerates CRLM by inducing hepatic inflammatory cytokines while reducing systemic natural killer and CD8+ T cell populations [24]. Similarly, Mohr et al. reported an increased hepatic expression of prometastatic factors (e.g., cytokines, chemokines, and adhesion molecules) during the advanced rate and burden of CRLM in alcohol-fed mice. Further, the processing of a known CRC prognostic indicator (carcinoembryonic antigen (CEA,)) by macrophages in the alcohol-affected liver was implicated in the production of the tumor-promoting factors [25]. In another study, computational modeling of an alcohol-permissive environment for CRLM showed the involvement of hepatic extracellular matrix (ECM) remodeling and macrophage polarization to M2-type tumor promoting cells [26]. 

The effect of alcohol on the metastatic behavior of CRC cells was evaluated in cell culture studies performed alone or in conjunction with CRLM assessments in animals or human tissue [17,18,22,27]. The results demonstrated that CRC cells exposed to ethanol displayed increased metastatic properties including migration, proliferation, and tumor cell survival. Moreover, unique alcohol-mediated pathways were identified including CRC migration induced through MCP-1 and GSK3β/β-catenin signaling [17] as well as an enhanced expression of CCL-5 leading to AMPK signaling [18]. Moreover, alcohol was found to promote CRC cell survival and metastatic potential through the activation of the Nrf2/HO-1 pathway [27] or the promotion of epithelial–mesenchymal transition induced via the TGF-β/RUNX3/Snail axis [22]. Overall, the mechanistic findings of the preclinical studies demonstrate a role of the alcohol-affected liver as well as the metastatic behavior of CRC cells to facilitate and promote CRLM (Figure 7). There were no major commonalities among the design of the studies and multiple mechanisms were uncovered. It is evident from the preclinical studies that CRLM is enhanced by alcohol intake and the identified mechanisms reveal potential targetable pathways in tumor cells and the host hepatic microenvironment to reduce the burden and poor outcomes associated with CRLM.

## 4. Discussion

Historical evidence indicates that alcoholic beverages have been part of the human diet for thousands of years, yet the relationship between alcohol use and cancer has only been investigated since the 20^th^ century [28]. Moreover, it was not until recent times that alcohol was officially classified as carcinogenic to humans by the International Agency for Research on Cancer [29]. Cancers of the colon and rectum were listed among the alcohol-associated diseases, with an enhanced risk of colorectal cancer reported for moderate to heavy alcohol consumption [30]. Although the evidence to date indicates that alcohol intake is an established risk factor for the development of primary colorectal cancers [31,32], the effect of alcohol use on CRC outcomes and contributing mechanisms remain to be defined. In this report, we performed a mapping review of studies conducted in the most recent decade that investigated the role of alcohol consumption and the advancement of CRC. In particular, our aim was to determine current research findings for the role of alcohol in CRC metastasis to the liver, the foremost site of secondary organ involvement and leading cause of CRC death [1,33]. Literature from 2010 to 2021 was reviewed to identify the effect of alcohol consumption on CRC outcomes. The search strategy retrieved 207 papers for initial review with 14 manuscripts fulfilling the defined criteria. Classifications of study type (preclinical or clinical in nature) as well as patterns in experimental designs and study results were compiled. 

The synthesis of findings from patient databases and alcohol use surveys showed both protective and harmful associations with an alcohol use history prior to CRC diagnosis. Poorer CRC prognosis was linked to increases in the quantity of alcohol consumed, while moderate wine intake was found to be suggestive of better outcomes. The progression of existent CRC and risk of CRLM was linked to heavy alcohol use. Sex-specific associations were observed in one-third of the clinical evaluations despite reports of higher CRC mortality and alcohol use disorders in males [34,35]. Limitations extracted in this review indicate that females were underrepresented due to the number of study participants or the lack of women that consumed alcohol in a specific population due to cultural and social issues [18,19]. Other noted limitations were related to variations in data collection methods, the alcohol concentration designation for heavy drinkers, and outcomes measured. Less than 25% of the studies evaluated the frequency (i.e., drinking days/week) of alcohol use in addition to the quantity consumed. The retrospective studies, while efficient for studying rare outcomes, were limited by the ability to establish a temporal relationship between alcohol exposure and CRC outcomes. All the clinical studies assessed alcohol consumption patterns prior to CRC diagnosis with many subjected to recall bias and reliance on baseline alcohol use data. Of note, self-reported alcohol data have been shown to be associated with an underestimation of alcohol consumption, especially in relationship to heavy intake patterns [36]. The limitations with self-reported data can also be compounded by cultural differences where the type of beverage, alcohol content, and drinking episodes can be highly variable [37]. To address research gaps, future evaluations are needed to fully characterize sex differences, cultural aspects, alcohol use patterns, and type of beverage consumed before and after initial diagnosis of colorectal cancer. Such investigations can lead to practice guidelines to address alcohol use by cancer patients and the potential for additional screening for suspected metastatic disease in those with a significant alcohol use history. 

Review of preclinical studies highlighted mechanistic findings for alcohol-enhanced CRC cell aggressiveness and liver metastatic disease. These works are critical in the understanding of potential targetable mechanisms considering the liver is the predominant site of alcohol-associated hepatic disease and CRC spread [3,33]. Prior research in this area showed that alcohol use correlates with synchronous as well as metachronous CRLM identified months after diagnosis [5,10]. The studies reviewed here show the most recent progress made in the identification of novel prometastatic pathways involved in the timing and extent of CRLM development in the alcohol-affected liver. Interestingly, the findings from each study present unique mechanisms highlighting the complexity of alcohol’s effects on the seeding and propagation of CRC cells in the host liver microenvironment. This information along with the association of alcohol intake with CRC mortality identified in the clinical studies [9,12,18,19,20,22] indicates the necessity to better define the complexity of CRLM for the development of therapeutics, especially for patients with a heavy alcohol use history. A limitation of the preclinical studies is the lack of translational research beyond the CRC cell culture studies and animal models. While animal models present many advantages, future research may benefit from human tissue analyses and xenografts. 

In summary, the current review of studies evaluating the relationship of alcohol consumption and the development of CRLM is timely and of clinical significance. The association of prevalent drinking habits with CRC outcomes is not fully understood and as reported here, is clearly understudied. The relevance of research in this area has been highlighted by recent reports concerning the potential impact of unprecedented drinking habits noted since the beginning of the COVID-19 pandemic. During the height of the shutdown period alone (March 2020 to September 2020), a 20% increase in alcohol sales was accompanied by excessive drinking patterns [6,38]. Unfortunately, the continued increase in alcohol consumption rates is expected to translate into significant health consequences in years to come with thousands of additional deaths estimated by 2040 that are due to alcohol-related liver disease and cancer [7]. It is difficult to assess what the unintended health consequences associated with alcohol consumption will be for patients that are/or will be diagnosed with CRC, but the need for continued and more extensive research is evident. 

## 5. Conclusions

This review highlights the importance of investigating the relationship between alcohol consumption and the development of colorectal liver metastasis. The findings indicate that during the most recent decade, a limited number of studies have been conducted concerning the role of alcohol consumption in CRC prognosis and CRLM. Although the studies found that alcohol use independently associates with negative CRC outcomes and enhanced liver metastatic disease, more work is needed to fully characterize involved mechanisms and potential therapeutic strategies. The research gaps identified include the unmet need for more in-depth prospective human studies along with preclinical assessments to better understand signaling pathways and cell populations that promote CRC metastasis in the alcohol-affected liver. Such evaluations are of high clinical importance considering the current healthcare concerns surrounding the prevalent use of alcohol and poor CRC outcomes. Future studies should include historical and longitudinal analyses of alcohol consumption patterns and outcomes in CRC patients, as well as detailed mechanistic evaluations for the identification of targetable pathways and the development of new therapeutic options.

## Figures and Tables

**Figure 1 biology-12-00257-f001:**
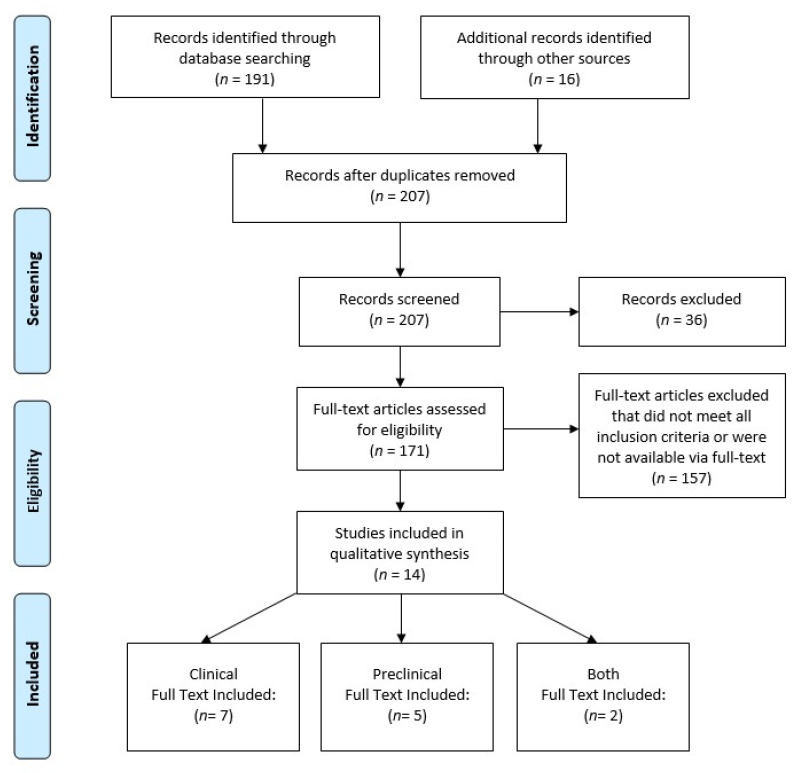
PRISMA flow chart showing the study selection paradigm. The additional sources used for record identification were obtained from bibliography screening of identified articles and applicable reviews. The database screening produced unique records and no duplicates were found. PRISMA diagram modified from Moher et al. [15].

**Figure 2 biology-12-00257-f002:**
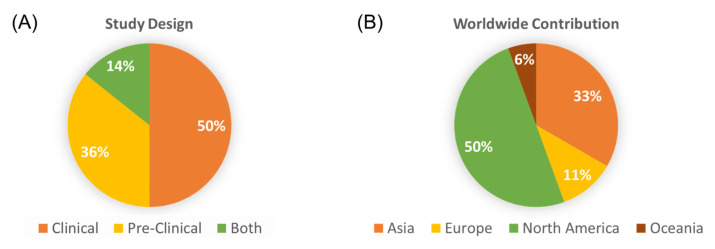
Study design and origin of contribution. (**A**) The percentage of studies according to the design (human subject data, pre-clinical, or a combination of both); (**B**) The distribution of studies according to the continent of origin.

**Figure 3 biology-12-00257-f003:**
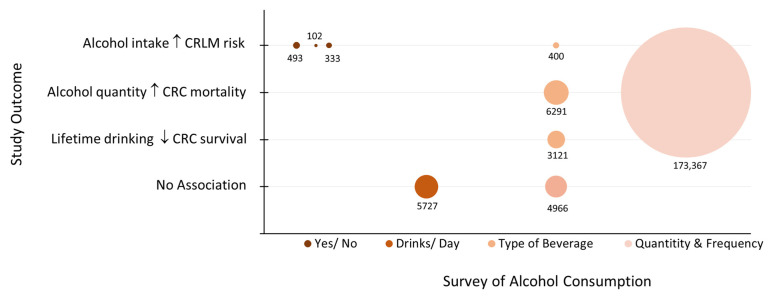
Bubble plot of clinical studies by outcome and survey of alcohol intake. The size of the bubble is proportional to the study sample size as noted. CRC: colorectal cancer; CRLM: colorectal liver metastasis.

**Figure 4 biology-12-00257-f004:**
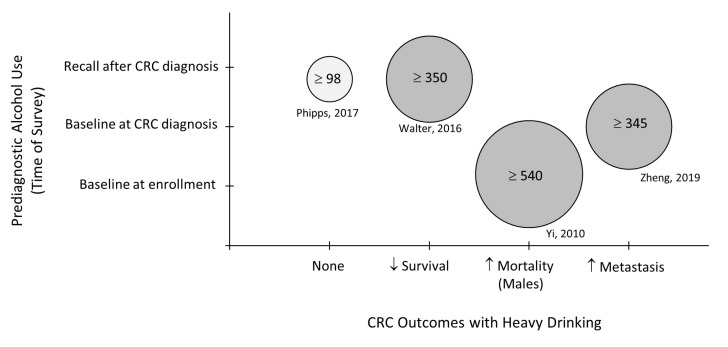
Bubble plot of studies reporting the type of alcohol beverage consumed by prediagnostic timeline and outcomes associated with heavy drinking. Bubble size reflects the noted quantity of alcohol (grams/week) that was designated as heavy drinking in each study [12,16,19,22]. CRC: colorectal cancer.

**Figure 5 biology-12-00257-f005:**
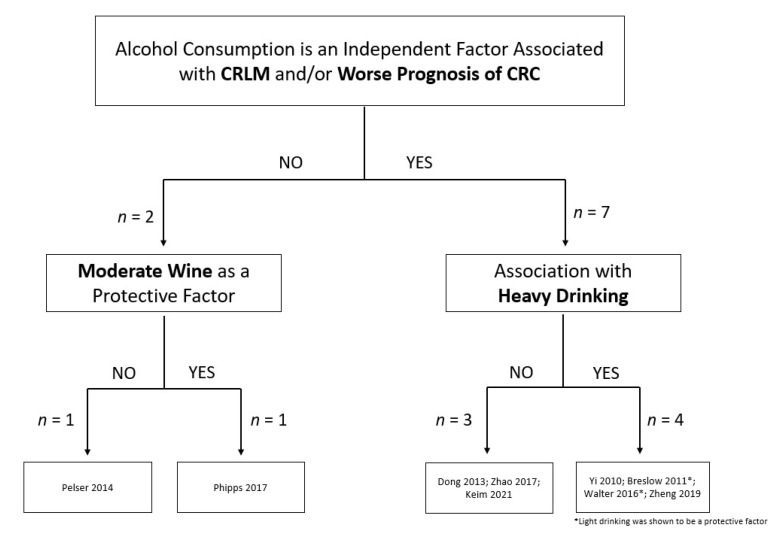
Summary of clinical reports investigating the association of alcohol and colorectal cancer outcomes [9,12,13,16,18,19,20,21,22].

**Figure 6 biology-12-00257-f006:**
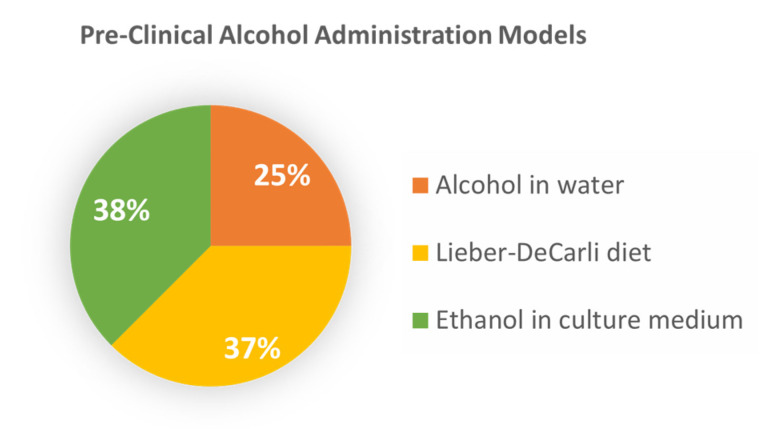
Alcohol exposure models used in preclinical studies. The percentage of studies is noted for the in vivo assessment of colorectal liver metastasis in animals fed alcohol in water or in Lieber–DeCarli diet; and colorectal cancer cell cultures treated in vitro with various concentrations of ethanol.

**Figure 7 biology-12-00257-f007:**
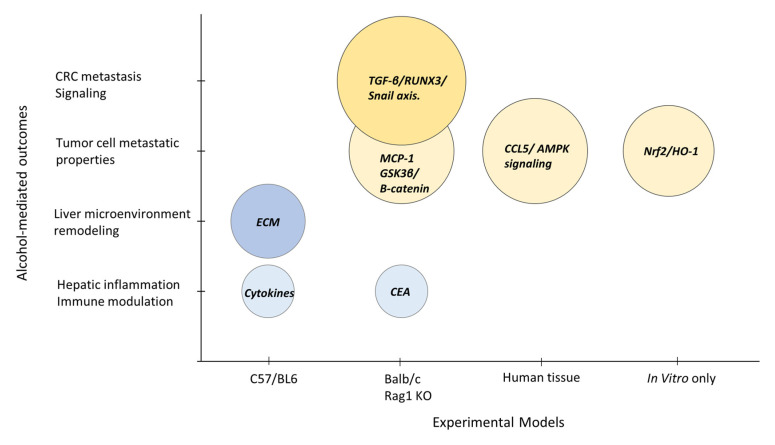
Summary of preclinical study results. Bubble plot reporting the study outcomes influenced by alcohol treatment in models using immunocompetent mice (C57/BL6) or immunodeficient mouse strains (Balb/c, Rag1 KO mice); human biopsy tissue; or in vitro only cell culture studies. The prometastatic pathways identified are highlighted for the alcohol-affected liver microenvironment (blue bubbles) or CRC tumor cell behavior (yellow bubbles). The bubble size reflects the number of tumor cells assessed in each study. CCL5: C-C chemokine ligand 5; CEA: carcinoembryonic antigen; CRC: colorectal cancer; ECM: extracellular matrix; MCP-1: monocyte chemoattractant protein-1.

**Table 1 biology-12-00257-t001:** Search criteria for alcohol and colorectal cancer (CRC) publications.

Inclusion Criteria	Exclusion Criteria
English language studies	Non-English
Studies from 2010 to 2021	Studies published prior to 2010
Alcohol	Reviews and meta-analyses
Alcohol-associated liver disease	Reports with overlapping data
Colorectal cancer mortality	Overall survival
Colorectal cancer liver metastasis	Exclusion of Stage IV CRC data
Clinical-based studies (prospective, retrospective, population studies)	No access to full text
Preclinical studies (animal, cell culture)	
Access to full text	

**Table 2 biology-12-00257-t002:** Characteristics of included clinical studies.

Reference	Country	Study Design	Subject (No.)	Sex (% Male)	Alcohol Use	Outcomes
Yi, 2010 [19]	Korea	Prospective Cohort	6291	42.8	Grams/week low to high Beverage type	Risk of CRC mortality postively associates with increasing alcohol consumption in men.
Breslow, 2011 [20]	USA	Prospective Pooled Survey Data	173,367 ^a^	52.3	Drinks/day Frequency Light to heavy	Higher-quantity drinking associates with increased risk of CRC mortality in women.
Dong, 2013 [9]	China	Retrospective	493	63.4	Yes/No	Alcohol use is an independent risk factor for CRLM.
Pelser, 2014 [13]	USA	Retrospective	5727	68.4	Drinks/day Moderate to Heavy	No association between alcohol intake and CRC mortality.
Walter, 2016 [12]	Germany	Prospective Cohort	3121	59.4	Grams/day Light to heavy Beverage type	Lifetime heavy drinking associates with poorer CRC survival.
Phipps, 2017 [16]	USA Canada, Australia	Prospective Pooled Survey Data	4966	52.0	Servings/week Beverage type	No associatation between beer or liquor consumption and CRC survival.
Zhao, 2017 [18]	USA, China	Retrospective	102	67.0	Yes/No	Alcohol use increases risk of advanced TNM stage, metastasis, and poorer prognosis.
Zheng, 2019 [22]	China	Retrospective	400	61.5	Grams/week Beverage type	Higher and frequent alcohol intake associates with CRC metastasis.
Keim, 2021 [21]	USA	Retrospective	333	58.0	Yes/No	Alcohol use in males associates with the risk of advanced CRC disease.

^a^ Subjects evaluated for colorectal cancer-specific outcomes. Abbreviations: CRC: colorectal cancer; CRLM: colorectal cancer liver metastasis; TNM: tumor, node, metastasis classification.

**Table 3 biology-12-00257-t003:** Characteristics of preclinical studies.

Reference	Experimental Model	Species/CRC Cell	Alcohol Treatment	Outcomes
Im, 2016 [24]	Animal	C57BL/6 mice MC38	Water/20% alcohol, 4–7 wks	Alcohol accelerates CRLM; alcohol altered hepatic microenvironment, inactivated immune surveillance.
Xu, 2016 [17]	Animal Cell culture	Balb/c mice HCT116, DLD-1, HT29, SW480	Water/2% alcohol, 4 wks 100–400 mg/dl ethanol	Alcohol increases the migration, metastasis of CRC cells; modulation of GSK3β/β-catenin/MCP-1 pathway.
Mohr, 2017 [25]	Animal	Rag1 KO mice LS174T	Lieber–DeCarli diet, 4–9 wks	Enhanced rate and burden of CRLM in alcohol-affected livers; CEA-mediated inflammatory mechanisms.
Zhao, 2017 [18]	Human tissue Cell culture	CRC biopsies HT29, DLD-1, RKO, SW480	Abstainers vs. drinkers 200 mg/dl ethanol	Alcohol use increases CCL5 expression in patient tumor tissue; CCL5 mediates CRC cell migration via autophagy and AMPK signaling.
Cernigliaro, 2019 [27]	Cell culture	HCT116, HT29, Caco-2	30–300 mM ethanol	Ethanol treatment leads to activation of Nrf2/HO-1 pathway; CRC cell survival and aggressive phenotype.
Hudson, 2019 [26]	Animal	C57BL/6 mice	Lieber–DeCarli diet, 6 wks	Modeling of M2 macrophages and ECM in metastasis.
Zheng, 2019 [22]	Animal Cell culture	Balb/c mice HT29, HCT116, LS174T, RKO, SW620,CT26	Lieber–DeCarli diet 100–200 mg/dl ethanol	Alcohol promotes CRLM via the TGF-β/RUNX3/Snail axis.

Abbreviations: CCL5: C-C chemokine ligand 5; CEA: carcinoembryonic antigen; CRC: colorectal cancer; CRLM: colorectal cancer liver metastasis; ECM: extracellular matrix; MCP-1: monocyte chemoattractant protein-1; Nrf2/HO-1: nuclear erythroid 2-related factor/heme oxygenase-1.

## Data Availability

The data presented in this study are available in Appendix A.

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
