# Peer review of "Alcohol Use and the Risk of Colorectal Liver Metastasis: A Systematic Mapping Review"

_biology, 2023, doi:10.3390/biology12020257_

Round 1
Reviewer 1 Report
The review manuscript by Roshan Sapkota and colleagues provides an overview of studies evaluating the association between alcohol consumption and colorectal cancer liver metastasis. This systematic review included literature on Pubmed and Embase from the year 2010 to 2021. A small number of studies reported that alcohol could promote colorectal cancer liver metastasis. The authors identified the research gap of the relationship between alcohol intake and CRC prognosis, and CRLM.
The manuscript is well-written, and the topic is interesting. I have some minor suggestions:
1. In figure 1 “additional records identified through other sources”. The authors need to explain in the figure legend what are the other sources.
2. Line 178 should be put after the figure 3 legend (lines 179 to 181)
3. Figure 7 summarizes the mechanistic pathway findings from the preclinical studies. I would recommend the authors make the bubble plot instead. The size of the bubble could reflect this pathway is identified/reported by how many studies.
Author Response
Reviewer 1
The manuscript is well-written, and the topic is interesting. I have some minor suggestions:
1. In figure 1 “additional records identified through other sources”. The authors need to explain in the figure legend what are the other sources.
Response 1: The legend for Fig. 1 has been edited to include the explanation of the other sources used for record identification.
2. Line 178 should be put after the figure 3 legend (lines 179 to 181)
Response 2: We have adjusted the formatting so the legend for Fig.3 is not separated from the graphic.
3. Figure 7 summarizes the mechanistic pathway findings from the preclinical studies. I would recommend the authors make the bubble plot instead. The size of the bubble could reflect this pathway is identified/reported by how many studies.
Response 3: As suggested, a bubble plot for Fig.7 has been made and the related text updated to include a description of the data presented in the new figure.
Reviewer 2 Report
Sapkota et. al., in their current review demonstrated the role of alcohol on the colorectal cancer. The authors study involved all the publications from 2010-2021 which makes the study comprehensive including the clinical and preclinical studies, which is the strongest part of the review. However, it needs more organization and some concerns need to be addressed as follows:
1. The abstract is very concise and should be reconstructed and rewritten by presenting a brief background on the topic, clarifying the aspects present in this review, conclusion, and further perspectives.
2. The mechanistic pathways involved should be discussed in more details.
3. The conclusion section needs to be rewritten in a more concise and informative way reflecting the main outputs of this review, practical applications, and future perspectives.
Author Response
Reviewer 2
Sapkota et. al., in their current review demonstrated the role of alcohol on the colorectal cancer. The authors study involved all the publications from 2010-2021 which makes the study comprehensive including the clinical and preclinical studies, which is the strongest part of the review. However, it needs more organization and some concerns need to be addressed as follows:
1. The abstract is very concise and should be reconstructed and rewritten by presenting a brief background on the topic, clarifying the aspects present in this review, conclusion, and further perspectives.
Response 1: The abstract has been rewritten as presented in lines 27-40.
2. The mechanistic pathways involved should be discussed in more details.
Response 2: Section 3.2 has been rewritten and contains more details of the mechanistic pathways as presented in lines 203-243.
3. The conclusion section needs to be rewritten in a more concise and informative way reflecting the main outputs of this review, practical applications, and future perspectives.
Response 3: The conclusion section has been rewritten (lines 343-357).
Reviewer 3 Report
In this literature review by Sapkota et al., the authors identify 14 research papers published between 2010 and 2021 that investigate the relationship between alcohol consumption and colorectal cancer progression, with special regard to CRC metastasis. Regardless of methodology and sample size, most papers have confirmed a relationship between alcohol consumption and CRC metastasis and/or poor outcome, and revealed potentially targetable cellular mechanisms. However, the authors argue that further research is needed to fill critical gaps in our understanding.
While initially I was reluctant to believe this topic was so understudied, a cursory PubMed search using a strategy similar to the one reported here verified that this was indeed the case. This underscores the bottom line of the paper, i.e., the need for prospective clinical studies to further evaluate the impact of alcohol on CRC clinical behavior and outcomes. The review is written in a succinct language; results are clearly presented; conclusions are solid and convey an important message. I only have very minor comments to add:
- In the last sentence of the Introduction, lines 78-80, the word “role” should probably be replaced by “relationship”: “…research investigating the relationship between alcohol consumption and…”
- The first sentence of subsection 2.1, “Search strategy”, lines 83-85, is slightly confusing. Consider rephrasing.
- In Table 1, capitals are used inconsistently (see phrase “Colorectal Cancer Mortality” vs. all others)
- Figure 1: Duplicate records must have already been removed in the first step, as 191 + 16 = 207
Author Response
Reviewer 3
The review is written in a succinct language; results are clearly presented; conclusions are solid and convey an important message. I only have very minor comments to add:
- In the last sentence of the Introduction, lines 78-80, the word “role” should probably be replaced by “relationship”: “…research investigating the relationship between alcohol consumption and…”
Response 1: The suggested change has been made as noted in lines 80-82 of the revised text.
- The first sentence of subsection 2.1, “Search strategy”, lines 83-85, is slightly confusing. Consider rephrasing.
Response 2: The phrase has been rewritten, see lines 85-87 of the revised text.
- In Table 1, capitals are used inconsistently (see phrase “Colorectal Cancer Mortality” vs. all others)
Response 3: The text in Table 1 has been corrected.
- Figure 1: Duplicate records must have already been removed in the first step, as 191 + 16 = 207
Response 4: Yes, the database screening produced unique records and no duplicates were found between those articles and the papers identified from other sources.